# Cloud drop activation of insoluble aerosols aided by film forming surfactants

Ari Laaksonen[1,2]

[1]Finnish Meteorological Institute, 00560 Helsinki, Finland
[2]University of Eastern Finland, Department of Technical Physics, 70211 Kuopio, Finland

**Correspondence:** Ari Laaksonen (ari.laaksonen@fmi.fi)

**Abstract.** Cloud droplet activation of insoluble aerosols covered by insoluble surfactant films has been studied theoretically by combining the FHH activation theory and an equation of state suitable for surfactant films that are in an expanded state. The key parameters governing the ability of the surfactant to suppress critical supersaturations are its partial molecular area at the water surface and the size of the molecule. For a fixed size, molecules with a larger molecular area are more efficient, whereas with a fixed area-to-volume ratio, smaller molecules are more efficient. Calculations made for stearic acid films on black carbon and illite aerosols indicate that the critical supersaturations are significantly lower than with pure particles, especially when the dry particle sizes are several hundred nanometers and larger. Furthermore, the reductions in critical supersaturation are similar when stearic acid is replaced by water-soluble organics with values of the hygroscopicity parameter ($\kappa$) up to 0.5. However, mixtures of surfactant and water-soluble organic compounds are relatively weaker at reducing critical supersaturations than either of these compounds alone which is caused by dilution of the surfactant film as the dissolved organic causes increased uptake of water vapor in the critical droplets. The theory has also been tested against experimental results on the impact of oleic acid films on the activation of calcite particles.

*Copyright statement.* TEXT

## 1 Introduction

The activation of cloud condensation nuclei (CCN) into cloud droplets was first theoretically described by Köhler in the 1920s (Köhler, 1926). The CCNs considered by Köhler were composed of hygroscopic salts, and in the atmosphere salt particles are certainly the most important CCN class. However, other types of aerosols, such as organic aerosols and mineral dust, can also act as CCN under suitable conditions. Furthermore, various physicochemical phenomena such as the co-condensation of soluble gases and surface tension depression by different types of surfactants can modify the cloud drop activation process. The impact of water-soluble surfactants has been studied extensively both theoretically and experimentally (e.g., Xiong et al., 2022; Vepsäläinen et al., 2023; Schmedding and Zuend, 2023; Bain et al., 2023; Laaksonen and Malila, 2021, and references therein). Less attention has been paid to water-insoluble film-forming surfactants. Brief theoretical considerations about their effects on

salt particle activation have been given by McFiggans et al. (2006) and Laaksonen and Malila (2021). Experimental studies have found a fairly minimal impact of insoluble surfactants on CCN activation of ammonium sulfate (Abbatt et al., 2005) and sea salt particles (Nguyen et al., 2017; Forestieri et al., 2018). Wang et al. (2018) studied experimentally the effect of oleic acid coatings on the activation of water-insoluble $CaCO_3$ particles. They observed reduced CCN activity with thin coatings, apparently because water was not able to penetrate between the core particles and the coatings in which the hydrophobic ends of oleic acid molecules were facing outward. With thick coatings, CCN activity increased relative to pure $CaCO_3$, but it was not clear how much this was due to the possible formation of oleic acid bilayers (with the hydrophilic ends of the molecules facing outward in the second layer) and how much due to surface-active film formation in the activating droplets. It appears that beside Wang et al. (2018), there are no other experimental or theoretical publications that would have examined the impact of film-forming surfactants on CCN activation of insoluble particles.

Small amounts of film-forming surfactants have a relatively insignificant effect on CCN activation of hygroscopic salt particles, for the simple reason that the activation diameters of cloud droplets are much larger than the dry CCN diameters. For example, the surface area of a 400 nm ammonium sulfate particle grows by a factor of 500 by the time it activates. This means that if only a few monolayers of insoluble surfactant cover the dry particle, the surfactant film becomes so dilute at activation that it has no effect on the surface tension of the droplet. With water-insoluble CCN the situation is different. The activation diameter of a droplet formed around a mineral particle is typically just a few tens of percent larger than the dry diameter, and therefore relatively small amounts of film-forming surfactants can have a much greater impact on CCN activation than with hygroscopic salts.

Atmospheric aging produces thousands of different organic substances, possibly also film-forming surfactants, on atmospheric aerosols, including water insoluble particles. Surfactants could also originate from primary sources such as oil spills. In addition to having possible atmospheric consequences, the influence of film-forming surfactants on the CCN properties of insoluble particles is of interest for fundamental aerosol studies as the problem has apparently not been treated theoretically before.

The purpose of this work is to theoretically examine the impact of water-insoluble surfactants on the CCN activation of black carbon (BC) and illite particles. These particle types were chosen as their CCN efficiencies are different (Kumar et al., 2011a; Laaksonen et al., 2020), making it possible to examine whether surfactants have proportionally similar impacts on their activation or not. The FHH (Frenkel-Halsey-Hill) activation theory (Sorjamaa and Laaksonen, 2007) is used to describe the interaction between water vapor and insoluble particles, and the reduction of surface tension of water by surfactant films is treated using the equation of state (EoS) derived by Gaines Jr. (1978). Stearic acid is chosen as the surfactant in most of the calculations as there is experimental data available to constrain the parameters of the EoS with pure water as the sub-phase of the stearic acid film (traditionally, the sub-phase water is almost always buffered to some specific pH, and the pH affects the isotherm - see e.g. Ha et al. (2000)). To be able to compare the theory against the CCN activation experiments of Wang et al. (2018), calculations are also made on the impact of oleic acid on activation $CaCO_3$ particles. Furthermore, the ability of film-forming surfactants to depress critical supersaturations of insoluble aerosols is compared to that of similar amounts of low hygroscopicity water-soluble organics that cause no surface tension depression.

## 2 Theory

The FHH adsorption activation theory combines the FHH adsorption isotherm (Frenkel, 1946; Halsey, 1948; Hill, 1949) and the Kelvin equation to express the equilibrium saturation ratio $S$ of water vapor on the surface of a spherical insoluble particle that has adsorbed $N$ monolayers of water as (Sorjamaa and Laaksonen, 2007)

$$S = \exp\left(-AN^{-B}\right)\exp\left(\frac{2\gamma_w v_w}{kTR}\right) \tag{1}$$

Here, the FHH parameters $A$ and $B$ describe the molecular interaction between the insoluble particle and water with $A$ representing the strength of the interaction and $B$ the rate of its decay as a function of distance. Increasing the value of $A$ and decreasing the value of $B$ both lead to increased adsorption. In the Kelvin term, $\gamma_w$ is surface tension and $v_w$ is molecular volume of water, $k$ is the Boltzmann constant, $T$ is temperature, and the droplet radius $R = R_d + d$ with $R_d$ the radius of the dry particle and $d = Nd_m$ the thickness of the adsorption layer. The thickness of a water monolayer $d_m$ is taken to be 2.84 Å.

Amphiphilic organics that are completely immiscible in water, such as long-chain fatty acids, may form so-called Langmuir films on the surface of water. The films are said to be in a compressed state when the film-forming molecules are close packed. Upon dilution, the films enter the so-called expanded state, and highly diluted films are said to be in a gaseous state because they obey the two-dimensional ideal gas law (Gaines Jr., 1978). Films in a gaseous state do not have much effect on the surface tension of water; in order to have a notable influence on cloud drop formation, the films should be in an expanded state at the point of activation. Gaines Jr. (1978) derived, starting from the Gibbs adsorption equation, an EoS that relates the surface area per surfactant molecule $\Omega_s$ - i.e. the variable representing the surface concentration of the surfactant molecules - and the surface pressure $\Pi = \gamma_w - \gamma$ that applies to expanded films. The surface tension $\gamma$ of the film can be calculated from the EoS:

$$\Pi = \frac{kT}{\omega_w}\left[\ln\left(1 + \frac{\omega_w}{\Omega_s - \omega_s}\right) - \ln f_w\right] \tag{2}$$

Here $\omega_w$ and $\omega_s$ denote the physical cross-sections of water and the surfactant at the surface, respectively. Like molecular volumes, the cross-sections are in principle partial molecular variables that depend on concentration; however, in the calculations below, they are taken to be constants. The volumes are assumed to be the same as their pure component values, and the cross-section for water, $\omega_w$, is taken to be 9.7 Å$^2$. When calculating the effect of the surfactant film on cloud drop activation, $\gamma_w$ in Eq.(1) is replaced by $\gamma$. Two surface pressure-surface area (Π-A) isotherms calculated using Eq. 2 are shown in Appendix A. It is easy to perceive the impact of the adjustable parameters $\omega_s$ and $f_w$ on the isotherms because the shape of the curve does not depend on them: changing the value of $\omega_s$ makes the isotherm move horizontally on the x-y plane, while changing the value of $f_w$ makes it move vertically (when $f_w = 1$, Π approaches zero asymptotically as $\Omega$ tends to infinity).

If the insoluble particle is coated, instead of film-forming surfactants, by a small amount of water-soluble material that does not affect the surface tension, then Eq. 1 is modified as follows (Kumar et al., 2011b):

$$S = \exp\left[-A\left(\frac{R - \epsilon_i^{1/3}R_d}{d_w}\right)^{-B}\right]\exp\left(\frac{2\gamma_w v_w}{kTR}\right)\exp\left(-\frac{\epsilon_s \kappa R_d^3}{R^3 - \epsilon_i R_d^3}\right) \tag{3}$$

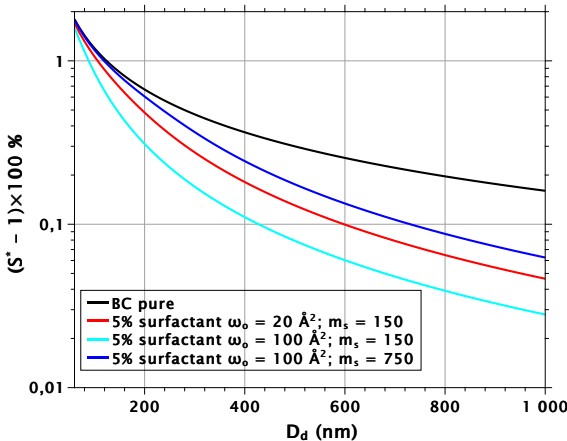

**Figure 1.** Critical supersaturations as a function of dry particle diameter for pure BC particles, and BC particles with 5% volume fraction of insoluble surfactant with cross-sections and molecular masses ($m_s$) as indicated in the legend. Density of the surfactant was assumed to be 850 $\mathrm{kgm^{-3}}$ in each case. Note that the combination of cross-section and mass may not be realistic especially in the most CCN active case shown here.

Here, $\epsilon_i$ and $\epsilon_s$ denote the volume fractions of insoluble and soluble material, respectively, in the dry particle, and $\kappa$ is the hygroscopicity parameter (Petters and Kreidenweis, 2007) of the soluble material. Gohil et al. (2022) have developed a model
for insoluble-soluble mixtures that accounts for, in addition to water adsorption, more complex mixture properties, such as solubilities of the dissolving species. However, Eq. (3) is sufficient for the purposes of the present work.

## 3   Results and discussion

### 3.1   The impacts of $\omega_s$ and $v_s$ on critical supersaturations

The three parameters that govern the ability of the film forming surfactants to impact cloud drop activation of insoluble particles
are the cross-section of the surfactant molecules $\omega_s$, the activity coefficient $f_w$, and the molecular volume $v_s$. The impact of the activity coefficient on cloud drop activation is straightforward; when its value is increased, the surface tension of water is reduced less, and the critical supersaturation ($S^*$) increases. Figure 1 shows schematically how $\omega_s$ and $v_s$ impact the CCN activation of BC particles (FHH parameters $A = 12$, $B = 1.93$; Laaksonen et al., 2020) with a 5% volume fraction of surfactant when the molecular mass is either 150 or 750 (density of the surfactant is assumed to be 850 $\mathrm{kgm^{-3}}$ in all cases) and the cross-
section is either 20 or 100 $\mathrm{\mathring{A}^2}$ (note that the combinations of these parameters used in the calculations do not refer to any specific surfactants and are not necessarily very realistic; furthermore, the activity coefficient is taken to be unity). As a base case, we have a model surfactant with low molecular mass and surface area (red line); critical supersaturations are shown as a

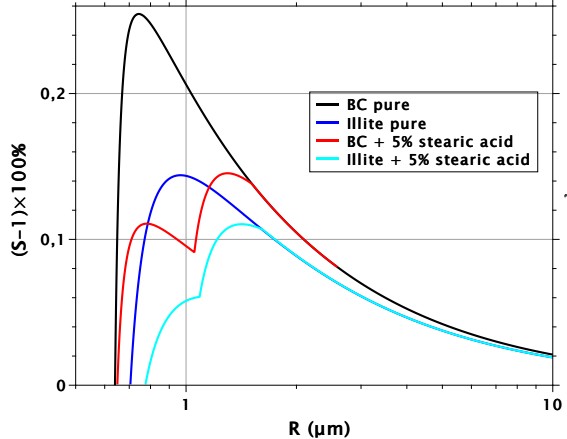

**Figure 2.** Köhler curves for pure BC and illite particles, and BC and illite particles with 5% volume fractions of stearic acid.

function of dry particle diameter between 60 and 1000 nm. Increasing the cross-section while keeping the molecular volume constant results in increased CCN activity (cyan line). However, if the ratio of the molecular volume of the surfactant to the cross-section is kept constant, and both the volume and the surface area are increased, the result is a decrease in CCN activity (blue line). The reason for this is that, at fixed surfactant volume fraction, the number of surfactant molecules in the system decreases with increasing molecular volume and therefore, the surface area per surfactant molecule $\Omega$ is larger at a given droplet size. Thus, with fixed molecular volume to cross-section ratio, the increase of $\Omega$ impacts the surface tension more than the increase of $\omega_s$, and as a result the critical supersaturation increases.

### 3.2 The impact of stearic acid films on CCN activation of BC and illite

Figure 2 shows Köhler curves for 600 nm BC and illite (FHH parameters $A = 1.02$, $B = 1.12$; Kumar et al., 2011a) particles, and their mixtures with a volume fraction of 5% of stearic acid. The activity coefficient and the cross-section of stearic acid were determined by adjusting them so that the theoretical $\Pi$-A-isotherm corresponds to the isotherm measured by Li et al. (2017) (Fig. A1).The curves for the mixed particles resemble those of CCN that contain sparingly soluble substances (Shulman et al., 1996) with double maxima in the BC curve and a kink in the illite curve. These features are caused by the limitation that the surface pressure cannot be higher than 38 mNm$^{-1}$ (see Appendix A), keeping the surface tension constant at a value of 34 mNm$^{-1}$ up to a surface area of about $\Omega_s = 30$ Å$^2$ per stearic acid molecule. The local minimum and the kink are formed at the points where the films reach this condition, and after further dilution, the surface tension increases until it reaches the value of pure water, and the Köhler curves of the mixtures unite with those of pure BC and illite, respectively. In the regime where the surface pressure is limited to a constant value, the film is thicker than a monolayer, and it is not certain what the correct value of the surface tension is. Furthermore, the Kelvin radius becomes somewhat ambiguous. On the one hand, the

radius should refer to the so-called surface of tension (Laaksonen and Malila, 2021), which is obviously in the top surfactant layer. On the other hand, the Kelvin equation describes the increase in vapor pressure of water due to droplet curvature, which forces the surface water molecules to be slightly further apart than on a flat surface. The outermost water molecules are below

the surfactant layer, and thus the radius at which they experience increased separation is somewhat smaller than the radius to the surface of tension, and it is not clear which radius should be used in the Kelvin equation (in the calculations shown here, the radius refers to that to the outermost water layer). The difference is small with expanded films, but in the constant surface pressure regime it is notable, and in cases where the first Köhler maximum is higher than the second maximum, the calculated critical supersaturations should be treated with caution.

Figure 3 shows the impact of stearic acid films on the critical supersaturations of BC particles as a function of dry particle diameter. The critical supersaturations were determined numerically with a simple procedure: the droplet diameter is increased incrementally, and the equilibrium saturation ratio is calculated after every increment. The critical supersaturation is reached when the equilibrium saturation ratio starts decreasing. To avoid possible local maxima such as that in the BC-stearic acid curve in Figure 2, the computation is started from a droplet large enough so that the surface pressure is just under 38 $\mathrm{mNm}^{-1}$.

However, in cases where the first maximum is higher than the second (see Section 3.3), the computation starts from a droplet that has adsorbed only a monolayer of water.

The black line in Figure 3 indicates pure BC, and the blue and red solid lines correspond to mixed particles with volume fractions 1% and 5% of stearic acid, respectively. Similarly as in Figure 1, the impact of the surfactant increases with increasing dry diameter. With the 5% volume fraction, stearic acid starts to impact the critical supersaturations at dry sizes of about 200

140    nm and larger, while with the 1% volume fraction, no impact is seen below 750 nm. The reason for this is that the surface-to-volume ratio decreases with particle size. Thus, the volume fraction 1% in a dry particle of 100 nm translates into a sparse surface coverage of stearic acid molecules on BC, with a single molecule occupying on average a surface area of 300 $\mathring{A}^2$. With a cross-section of 25 $\mathring{A}^2$, the complete monolayer coverage of the BC particle occurs only close to $D_d$ = 1200 nm. With a volume fraction 5%, stearic acid completely covers the BC particle at $D_d$= 240 nm.

The dashed lines in Figure 3 show critical supersaturations for BC particles mixed with 1 and 5% volume fractions of water-soluble organic with $\kappa = 0.05$ (corresponding to organic species with molecular masses on the order of 300 - 350, such as fulvic acids (Petters and Kreidenweis, 2007)). In the calculations, it was assumed that the organic species has no effect on water surface tension. With volume fractions of 5%, water-soluble organic and stearic acid suppress the critical supersaturations of BC particles by almost the same amount at dry diameters larger than about 400 nm. However, with volume fractions of 1%,

the water-soluble organic is more efficient than the surfactant.

The solid cyan line in Figure 3 shows critical supersaturations for BC particles mixed with 2.5% volume fraction of stearic acid and a similar volume fraction of the organic with $\kappa = 0.05$. Interestingly, this mixture shows reduced CCN activity compared with either of the 5% volume fraction mixtures. This is especially important to note, as it can be difficult to produce particles coated with pure insoluble surfactant through the vapor deposition mechanism so that water can penetrate through the

coating during a CCN activation experiment (Schwier et al., 2012; Wang et al., 2018). More practical methods to ensure film

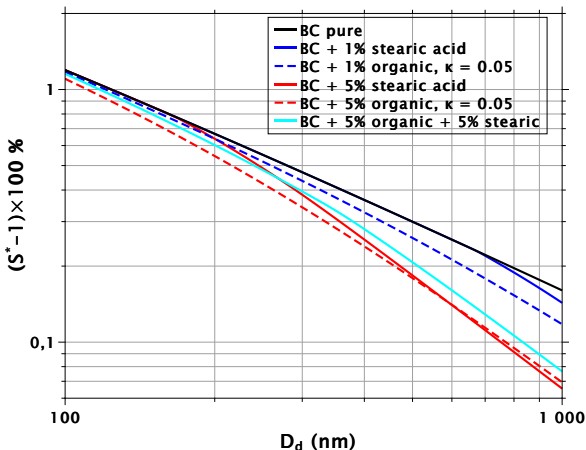

**Figure 3.** Critical supersaturations as a function of dry particle diameter for pure BC particles, BC particles with 1% and 5% volume fractions of insoluble surfactant (stearic acid), BC particles with 1% and 5% volume fractions of soluble organic with $\kappa = 0.05$, and BC particles with 2.5% volume fractions of both stearic acid and the organic with $\kappa = 0.05$.

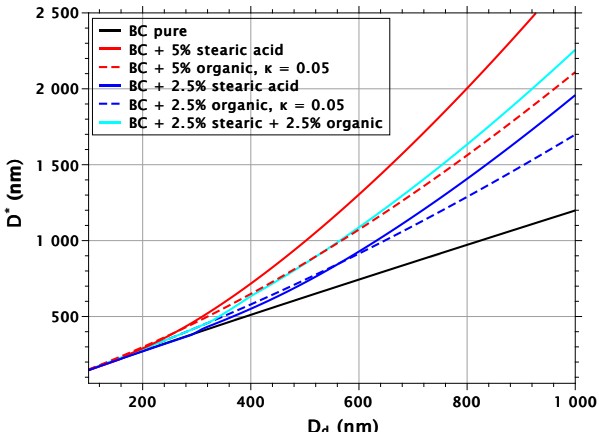

**Figure 4.** Critical radii as a function of dry particle diameter for pure BC particles, BC particles with 5% and 2.5% volume fractions of stearic acid, BC particles with 5% and 2.5% volume fractions of water-soluble organic with $\kappa = 0.05$, and BC particles with 2.5% volume fractions of both stearic acid, and the organic with $\kappa = 0.05$.

formation on the activating cloud droplet might involve introducing some water-soluble species together with the insoluble surfactant. It would then be important to be able to account for the joint effects of all species on CCN activation.

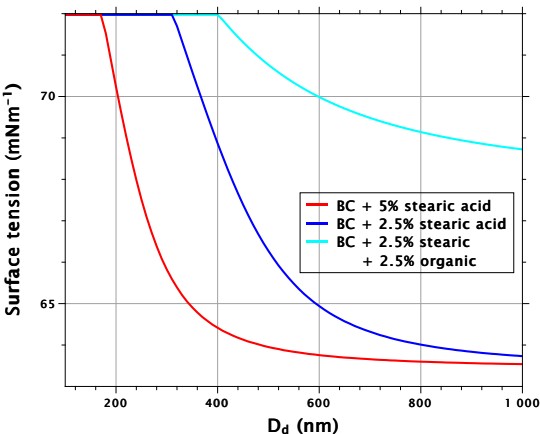

**Figure 5.** Surface tension of critical droplet as a function of dry particle diameter for BC particles with 5% and 2.5% volume fractions of stearic acid, and BC particles with 2.5% volume fractions of both stearic acid and water-soluble organic with $\kappa = 0.05$.

The reason for the non-additivity of the effects from surfactant and soluble organic can be understood by considering the sizes of critical nuclei, and their surface tensions. Figure 4 shows the critical radii of BC particles and their mixtures with stearic

acid and organic with $\kappa = 0.05$. With the 5% mixtures (red lines), the difference between stearic acid and the water-soluble organic substance is greater than with the critical supersaturation. Similarly as with water-soluble surfactants, reduced surface tension leads to increased critical radii compared to water-soluble species that reduce critical supersaturation without affecting surface tension (Ruehl et al., 2016). The same is true for the 2.5% mixtures above the dry diameter of 600 nm. An important thing to note is that the 2. 5% mixture of organic water-soluble and BC has up to 40% larger critical diameter than pure BC.

Therefore, when 2.5% stearic acid is added to the droplet formed on a mixed organic-BC-particle, the stearic acid film will be considerably more dilute than when the same volume fraction is added to a droplet formed on pure BC. Indeed, Figure 5 shows that the reduction in surface tension in critical droplets is much lower with the stearic acid-organic-BC particles than with the stearic acid-BC particles, explaining the non-additivity of the effects from the surfactant and the water-soluble organic.

Figure 6 is similar to Figure 2 otherwise, but the insoluble particle is illite, which is more CCN active than BC. Stearic acid

does not suppress critical supersaturations as effectively with illite as with BC; with 1-micron particles, $(S^* - 1)$ of illite drops by 39% due to 5% volume fraction of stearic acid, while with BC the drop is 59%. In addition, a volume fraction 2% of stearic acid is needed to produce a similar effect with illite as a volume fraction 1% produces with BC. Furthermore, with illite, the water soluble organic with $\kappa = 0.05$ causes a slightly greater reduction of critical supersaturation relative to stearic acid, than with BC. Because illite is more hydrophilic than BC and adsorbs more water, the stearic acid films on droplets formed around

illite are more dilute and reduce the surface tension of water less than the films covering droplets formed on BC. This explains why stearic acid suppresses critical supersaturations more efficiently with BC than with illite.

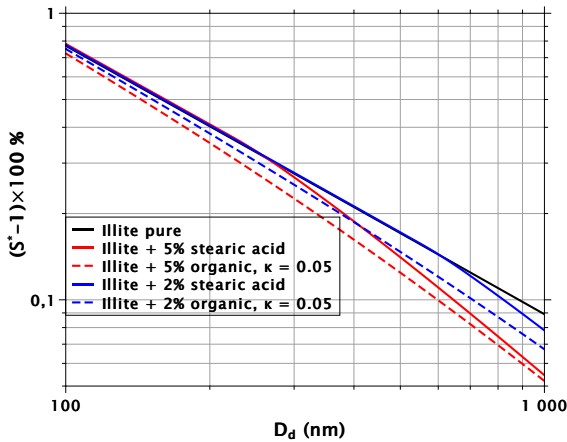

**Figure 6.** Critical supersaturations as a function of dry particle diameter for pure illite particles, illite particles with 2% and 5% volume fractions of stearic acid, and illite particles with 2% and 5% volume fractions of soluble organic with $\kappa = 0.05$.

### 3.3 Comparison of theory and experiments on CCN activation of oleic acid coated CaCO$_2$ particles

Figure 7 shows a comparison of the theory and experiments of Wang et al. (2018) who measured the activation of pure calcite particles and calcite particles coated with oleic acid. The black spheres and the black line show experiments and theory, respectively, for uncoated particles. The theoretical line was obtained using FHH parameters $A = 0.5$ and $B = 1.08$. As can be seen, the match with the experiments is not quite perfect, but is good enough for the purposes of the present exercise (the slope of the theoretical line could be improved slightly, but that would involve setting $B$ at an unrealistically low value).

The blue crosses show critical supersaturations for calcite particles coated with 4.3% volume fractions of oleic acid. Wang et al. speculated that the increased critical supersaturations with particles coated with low volume fractions of oleic acid might be explained by oleic acid not letting water through to the calcite surface. Instead, the hydrophilic ends of oleic acid molecules would stick to the surface of the calcite, leaving the hydrophobic ends facing outward, and making the particles more hydrophobic compared to pure calcite. However, the smallest (200 nm) particles have the lowest surface coverage, but their critical supersaturation is elevated more than that of the largest (300 nm). particles, which have the highest surface coverage of oleic acids in the scenario of Wang et al. Indeed, if theory is fitted to the data points assuming that no oleic acid resides at the surface of the water, the theoretical slope is even worse compared to the experimental slope than with the pure particles.

The blue theoretical line, which fits the experiments quite well, was obtained assuming that only a fraction of the oleic acid (corresponding to a surface coverage of 130 Å$^2$ per molecule) sticks to the surface as Wang et al. suggested, and the rest migrates to the surface of the droplets. The FHH parameters used in the calculation were $A = 0.2$ and $B = 1.6$. The slope of the theoretical line reflects the impact of surface tension: with the smallest particles, there is not enough oleic acid on the

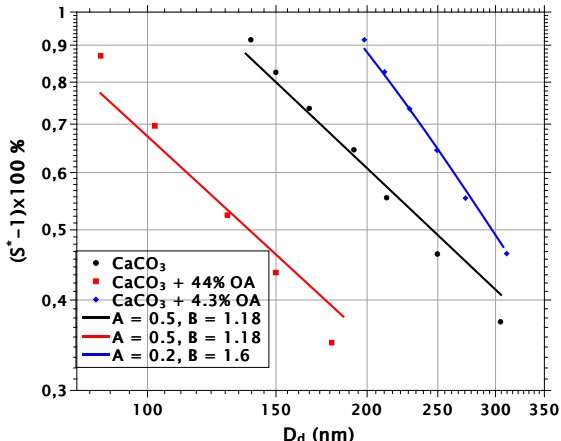

**Figure 7.** Experimental (Wang et al., 2018) and theoretical critical supersaturations as a function of dry particle diameter for pure and oleic acid (OA) coated calcite particles.

surface of critical droplets to reduce the surface tension, whereas in the large-particle end of the line, the surface tension of critical droplets is reduced.

The red squares show experimental critical supersaturations at 44% volume fraction of oleic acid. If a scenario consistent with the blue theory line is assumed, the theory does not reproduce the observed critical supersaturations, as the prediction (not shown) is close to that for uncoated particles. The cause of this is not the assumption that a small amount of oleic acid remains on the calcite surface; rather, the FHH parameters $A = 0.2, B = 1.6$ make the particles so hydrophobic that the reduction of surface tension by oleic acid is not sufficient to bring the theory in agreement with experiments. However, if it is assumed that the FHH parameters are those of the pure calcite particles, i.e. that in this case no oleic acid molecules are stuck to the calcite surfaces, the red theoretical line is obtained. As with pure particles, the slope of the line is somewhat different from the experiments, but clearly the decrease of critical supersaturations is of correct magnitude.

In general, the comparison between theory and experiments is ambiguous. The theory explains both the slope of the thinly coated calcite particles and the magnitude of the reduction of critical supersaturation of the thickly coated particles. However, the theoretical scenarios in the two cases are discordant. It is unclear why in the former case a fraction of oleic acid molecules would stick to the surfaces of the calcite particles, making them more hydrophobic, but not in the latter case. Furthermore, it should be noted that with particles that contain the 44% volume fraction of oleic acid, the first maximum in the Köhler curve is higher than the second, thus defining the critical supersaturation. As discussed in Subsection 3.2, the first Köhler maxima occur when the droplets are covered by more than a monolayer of surfactant, which makes theoretical predictions rather uncertain.

## 4 Conclusions

The impact of water-insoluble film-forming surfactants on the cloud drop activation of insoluble particles was studied theoretically by combining the FHH activation theory and an EoS for expanded surfactant films. It appears that this subject matter has not been studied theoretically before. The surfactant properties that influence critical supersaturations are the partial molecular area and the partial molecular volume. Increasing the area while keeping the size of the molecule constant leads to decreased critical supersaturations; however, if the ratio of the molecular area and volume is kept constant, smaller molecules are more efficient in promoting CCN activation. In addition, the activity coefficient of water in the surface film impacts the surface tension and thereby also the critical supersaturations.

Model calculations were made for systems consisting of BC or illite particles with stearic acid as the film-forming surfactant. In addition, the impacts of the surfactant on critical supersaturations were compared to those of an unspecified organic compound with a hygroscopicity parameter $\kappa = 0.05$. A volume fraction of 5% stearic acid suppressed critical supersaturations of BC and illite particles with quite considerable effects at dry particle diameters of about 300 nm and larger. The percentage impact on the critical supersaturation was greater with BC particles than with illite. With lower volume fractions, the suppression of critical supersaturations was smaller but not completely insignificant at large particle diameters close to a micrometer.

Compared to the water-soluble organic, the impacts of stearic acid on critical supersaturations were quite similar when BC particles and 5% volume fractions of stearic acid or the organic were considered. In all other mixtures, the water-soluble organic was more CCN efficient. Interestingly, it was observed that the impacts of stearic acid and the water-soluble organic on the CCN activity of insoluble particles are not additive, as a mixture with 2.5% volume fractions of both substances reduced the critical supersaturations of BC particles less than in either of the 5% volume fraction cases. This can be explained by the fact that the water-soluble organic material causes increased uptake of water vapor compared to pure BC particles, which dilutes the stearic acid film efficiently, and therefore reduces the surface tension effect on CCN activation. For the same reason, stearic acid suppresses the critical supersaturation less with illite than with BC: illite adsorbs more water and hence has more dilute stearic acid film than BC. In general, film-forming surfactants tend to aid CCN activation more with more hydrophobic particles. However, it should be kept in mind that dissolved species can affect the surface pressure-surface area isotherm of the surfactant (an effect that was not considered in this work) and thus counteract this tendency to some extent.

The theory was also tested against experimental results of the impact of oleic acid on CCN activation of calcite particles. The theory appears to explain both the increased CCN activation with thick oleic acid coatings and the decreased CCN activation with thin coatings; however, the assumptions of the interaction of oleic acid with the calcite surface were contradictory in these two cases, and therefore some ambiguity remains. Furthermore, with the thick coatings, the theory predicts that the critical droplet sizes are so small that there is more than a compressed monolayer of oleic acid on the droplet surface, which makes both the exact value of surface tension and the definition of the droplet radius to be used in the Kelvin equation somewhat uncertain. The theoretical predictions made here should therefore be treated with some caution. If future lab studies are conducted on the impacts of film-forming surfactants on CCN activation of insoluble particles, it would be advisable to use coatings thin enough so that the films on activating droplets are expected to be in an expanded state.

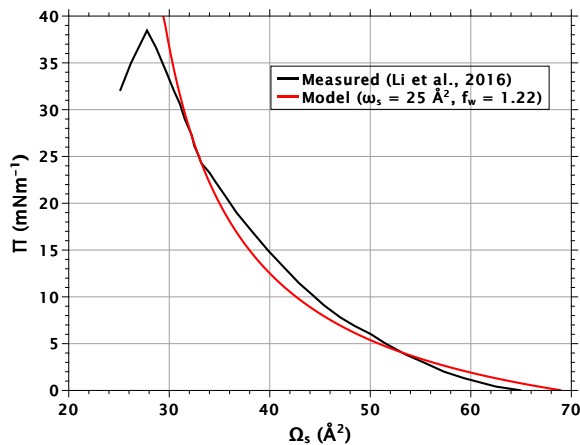

**Figure A1.** Measured and modelled surface pressure - area isotherms for stearic acid. The subphase for the measurement curve was pure water.

*Code availability.* The FORTRAN codes used for the calculations are available upon request to Ari Laaksonen (ari.laaksonen@fmi.fi).

## Appendix A: Pi-A-isotherms for stearic and oleic acids

Figure A1 shows measured Li et al. (2017) and modeled $\Pi$-A-isotherms for stearic acid on pure water. The measured pressure shows a maximum at $\Omega_s$ of about 28 Å$^2$. Here, the compressed monolayer film collapses into three-dimensional structures. In theoretical calculations, the film pressure is therefore limited to a maximum of 38 mNm$^{-1}$. Figure A2 shows experimental and modeled isotherms for oleic acid. Note that the experimental isotherm is buffered to pH = 6. Ha et al. (2000) measured isotherms at four different pH values. In this work, the isotherm measured at pH = 6 was selected because it corresponds better to atmospheric droplets whose pH is reduced by absorbed carbon dioxide than the isotherm measured at pH = 7. Ha et al. (2000) used HCl and NaOH to buffer the water and stated that "at specific pHs the isotherms were insensitive to the amount of HCl or NaOH added". With oleic acid, the film pressure is limited to a maximum of 30 mNm$^{-1}$ in theoretical calculations.

*Competing interests.* The author declares no competing interests.

*Acknowledgements.* This work was supported by the Research Council of Finland Flagship ACCC (grant no. 337552) and MEDICEN project (grants no. 336557 and 345125).

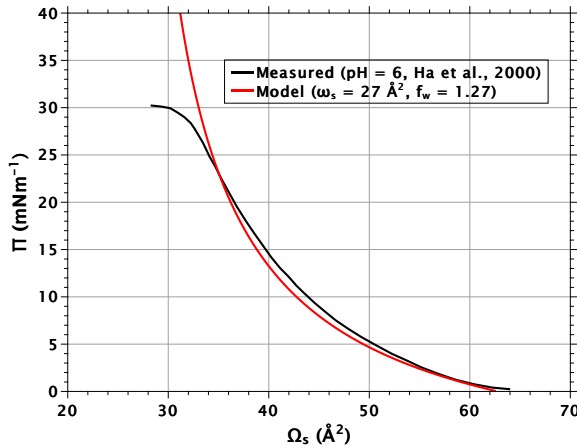

**Figure A2.** Measured and modelled surface pressure - area isotherms for oleic acid. The subphase for the measurement curve was water buffered to pH = 6.

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
