# Peer review of "Cloud drop activation of insoluble aerosols aided by film forming surfactants"

_Aerosol Research, 2024_

## Author Response (AR1)

I thank both reviewers for many excellent comments, which have helped to improve the manuscript considerably. As detailed in my replies to Reviewer 1, I found that the stearic acid Pi-A-isotherm that I used in my original calculations does not correspond to the experimental isotherm measured using pure water, but an aqueous aluminum chloride solution. Correcting this mistake led to somewhat different results and conclusions compared to the original manuscript.

**Reviewer 1:**

**This study investigates the influence of surfactant films on the activation of insoluble aerosol particles with a theoretical modelling approach. The model is introduced and in the following applied to black carbon and illite particles coated with either stearic acid or a water soluble compound. Sensitivity of the results to surfactant properties is shown as well. The study presents two interesting findings: First, a coating with insoluble surfactants can facilitate CCN activation, and second, the activation enhancement of insoluble surfactants and water soluble organics seem not to add up linearly. The paper presents the results in a clear and concise way, however, it lacks some fundamental information about the (atmospheric) relevance of the general topic and the chosen cases. Furthermore, it is not clear what the novelty of the study is and therefore, it is not possible to judge if the study provides a substantial contribution to scientific progress. The paper would also benefit from placing the results in the context of the literature, from discussing the results in more detail, and from discussing uncertainties and limitations of the model. My specific comments to the individual sections are:**

1. **The introduction describes why insoluble surfactant films can have a substantial impact on insoluble droplet activation (line 30-35), but it is not mentioned**
   - **why a change in droplet activation is of scientific interest, especially the activation of insoluble aerosol particles?**
   - **Where do insoluble particles with films of insoluble surfactants occur?**
   - **How widespread are they?**
   - **What are sources of insoluble surfactants in the atmosphere?**
   - **Why are BC and illite chosen in this study?**
   - **Why stearic acid?**
   - **Why is not CaCO3 chosen, which would allow a comparison to the experimental results by Wang et al. 2018?**
   - **What is the novelty of the study?**

   I have added text to the introduction (lines 30-32; 41-45; 51-55) to discuss these questions. Atmospheric aging produces different types of organic substances, possibly also film-forming surfactants, on atmospheric aerosols – including water insoluble species. The question regarding the prevalence of atmospheric mineral (or soot)-surfactant mixtures is of course interesting, but something that is very hard to answer and beyond the scope of this paper, which aims at providing a theoretical framework for further studies. I believe that this work also has relevance to fundamental aerosol science. Regarding the choice of BC and illite: as pure species there is a clear difference in their CCN properties, and I wanted to investigate whether the surfactants have proportionally similar impacts on their activation or not. Stearic acid was chosen because there is Pi-A-isotherm data for stearic acid with pure water as a subphase (see below for some more elaboration), which is not the case with most surfactants. Traditionally, the sub-phase water is buffered to maintain a constant pH, and this buffering can impact the isotherms. Choosing CaCO3 as one of the CCN species is not

sufficient alone to compare to Wang et al as they used oleic acid as surfactant, and I have not found Pi-A-data for oleic acid with pure water as a sub-phase. However, I have now added calculations which can be compared to Wang et al. (new subsection 3.3), with the assumption that data with water buffered to pH = 6 is sufficiently reliable to use as a basis for modelling the surface tension.

What comes to novelty of the study: I have not found any papers in the literature that would have presented a theoretical study of the impact of film-forming surfactants on the CCN activation of insoluble particles. This of course does not prove that such studies do not exist. However, I have now addressed the question in the Introduction.

2. **Theory:**
   o **Line 47: Help the reader by describing what A and B do qualitatively, like: "A low value of A refers to a strong interaction, i.e., a rather hydrophilic surface", or vice versa. See also line 104: "Illite is somewhat more CCN active than BC" – can this be seen from the lower A parameter?**

   Done.

   o **Line 52: "amphiphilic organics […] are completely immiscible in water…": I am wondering whether the acidic end of a fatty acid can interact with and partially dissolve in water. The presented model only takes into account the effect on the surface tension, but can the acidic group also change the water vapor pressure by some other (adsorption) effect? If this is an uncertainty of the model, it should briefly be discussed and estimated what effect it could have.**

   If the surfactant molecule does not submerge completely into the bulk but remains part of the surface phase, then it will not have any effect on water activity nor on water vapor pressure. It is possible that the surfactants do have an impact on sticking probability (or accommodation coefficient) of incoming water vapor molecules, which may subsequently have an effect on cloud activation, but this is a non-equilibrium effect which slows down the approach from a non-equilibrium state to equilibrium that cannot be addressed within a strictly equilibrium theory such as the one considered in this work; rather, a cloud model approach would be needed.

   o **Line 53-55: A reference should be given to the theory of a compressed, expanded and gaseous state and the two-dimensional ideal gas law.**

   Done (line 71).

   o **Line 58 and 61: What is the difference between "the surface area per surfactant molecule" ($\Omega_s$) and the "partial molecular surface area of […] the surfactant" ($\omega_s$)?**

   The former is total surface area divided by the number of surfactant molecules, whereas the latter is the physical cross-section of a single surfactant molecule on the surface. This has now been clarified in the text (lines 74 and 77-78), and I now use the term cross-section instead of molecular area when discussing $\omega_s$.

- How does equation 2 depend on the dilution/composition of the surface film or the adsorbed water mass or the wet diameter? Is it via $\Omega_s$ and if so how? The specific equations should definitely be given, especially since the model code is not directly accessible.

  As noted above, $\Omega_s$ is total surface area divided by the number of surfactant molecules, which of course grows with the second power of the wet diameter. This has now been clarified (line 74).

- In the gaseous state following the 2D ideal gas law it is understandable that the water activity coefficient is approximated by unity. However, Line 64: "The film is assumed to be dilute enough …" seems to be in contrast with "that applies to expanded films." in line 59. If the surface film is in an expanded film state, how can it be ensured that $f_w=1$ is justified? It should be discussed what uncertainties this assumption introduces, at least qualitatively.

  This is a very good comment, and one that prompted me to check my equations against experimental data once more. It turns out that I have made an unfortunate error when verifying the theoretical Pi-A-isotherm for stearic acid: instead of comparing to data measured using pure water as sub-phase, I have accidentally used a data file that corresponds to a Pi-A-isotherm with aqueous $AlCl_3$ solution as sub-phase (Li et al., 2016). I have now corrected for the error, and the correction has led to a number of changes in the conclusions, the most important ones being that 1% of stearic acid has almost no effect on CCN activation, and 5% of stearic acid impacts CCN activation only when dry particle size is larger than 200 nm. Furthermore, at large dry sizes, the impact of 5% stearic acid is somewhat reduced so that above about 400 nm dry diameter, the effect is similar to water-soluble organic with kappa = 0.05 rather than 0.09. In the new stearic acid model the activity coefficient is 1.22, and the impact of activity coefficient larger than unity (reduction of critical supersaturation) is mentioned on lines 95-97.

  I have also added comparisons of the theoretical and experimental Pi-A-isotherms of stearic and oleic acid to Appendix A.

  S.Li, L. Du, Z. Wei, W. Wang: Aqueous-phase aerosols on the air-water interface: Response of fatty acid Langmuir monolayers to atmospheric inorganic ions. Sci. Total Environ. 580, 1155-1161, 2017

3. **Results:**
   - Line 72: "The two parameters that govern…": How does S depend on $v_s$? The mathematical relationship should be given or at least qualitatively be described.

     For a constant volume fraction of surfactant, the number of surfactant molecules increases with decreasing $v_s$. The surface tension is a function of $\Omega_s$, i.e. the total surface area of a droplet divided by the number of surfactant molecules. The more there are surfactant molecules, the lower the surface tension. Thus, larger $v_s$ => higher surface tension at a given drop size => higher critical supersaturation. This explanation has now been added (lines 106-109).

o   Before showing results of the critical supersaturation in Figure 1, it should be explained how the critical supersaturation is determined. Also it would be extremely useful to show in a plot the development of S over $d_{wet}$ (Köhler curve) or on a similar x-axis and how the surface tension, the film thickness evolve during water uptake. This would help to understand how S* was derived and help to illustrate other processes that are discussed in the following in the text.

The numerical procedure is now described in the text (lines 131-136), and Köhler curves + discussion have been added (Fig 2 and lines 111-129).

o   Line 75: Is a density of 850 kg/m$^3$ realistic? Any reference? (Wikipedia gives a density for stearic acid of 941 kg/m$^3$)

Figure 1 is meant to show schematically how the critical supersaturations depend on the molecular parameters of the surfactant, rather than to predict the critical supersaturations for some specific systems. Changing the density from 850 (which happens to be the density of phytol, another film-forming surfactant) to 941 would just shift the curves slightly.

Line 79-81: "However, if the ratio of surfactant molecular volume to surface area is kept constant, and both the volume and the surface area are increased, the result is decreased CCN activity (blue line)" – Why? Can this be explained on a molecular level?

The reason is that the difference ($\Omega_s - \omega_s$) in Eq. (2) increases when the molecular volume and area are increased and their ratio is kept constant, which makes the surface pressure lower and surface tension higher. See explanation on lines 106-109.

o   Line 91-92: "Water soluble organic with $\kappa=0.09$" – what molar mass does this approximately correspond to (assuming ideality and no dissociation)? Can an example be given for an organic compound with approx. $\kappa=0.09$?

As explained above, the correct kappa value is actually 0.05 (when comparing the impact of soluble organic to that of stearic acid). The molar mass would be on the order of 300-350. Examples of compounds having kappa values of about 0.05 are fulvic acids (Petters and Kreidenweis, 2007) (added on lines 146-147).

o   Line 97-98: "Interestingly, this mixture shows markedly reduced CCN activity compared with either of the 5% volume fraction mixtures." – since this is a major new finding in this study, it should be elaborated more on it. Why could it be that it behaves like that? What was expected? How does it compare to a case with 2.5% water soluble & 0% stearic acid or to 2.5% stearic acid & 0% water soluble? Is there at least an enhanced activation compared to one of those cases?

The reason for this is basically the same as why the surfactants impact insoluble aerosol activation but not the activation of soluble particles. Addition of water soluble organic to the insoluble particle increases its uptake of water vapor. The

droplets become larger, and the surfactant films more dilute, reducing the surface tension effect on activation. Discussion has been added on lines 158-159, 163-168, and 230-236.

- o **Line 98-99: "This is especially important to notice, as it may be difficult to produce particles coated with pure insoluble surfactant via vapor deposition mechanism" - Here also a reference to the atmospheric situation could be made since it is rather unlikely to have pure surfactant films of one single compound on atmospheric aerosol particles, but rather organics appear in complex mixtures. This is also something that should be discussed. Are the pure surfactant films in this study even realistic? What could be different for more complex compositions?**

  Indeed, pure surfactant films are unlikely in the atmosphere except maybe in rare situations where some major leak of oil occurs (such as the Deepwater Horizon case). However, pure surfactant films can be made in the lab to test the theory. Obviously, more complex situations than e.g. the 50/50 mixture of insoluble surfactant and water-soluble non-surfactant can be dealt with by adding complexity to the model; however, in atmospheric situations the problem is often such that even if the mass spectrum of the organic mix is known. it is difficult or even impossible to relate to a specific mixture of well-defined compounds. In any case, the purpose of this paper is to lay out the theoretical treatment of the impact of film forming substances on the activation of insoluble particles, and therefore I have left out further speculation of what may be going on with atmospheric mixtures.

- o **Line 114: I assume "the surface tension drops rapidly" should be "the surface tension 'at activation' drops rapidly"**

  That is correct. However, the sentence has now been removed.

- o **Since the model is not validated by experimental data, the uncertainties of the model approach should be discussed and quantified.**

  The model is now compared against the oleic acid data of Wang et al. (2018), and uncertainties are discussed in that context. New subsection 3.3.

- o **Finally, how do the results compare to previous work (if there is some)?**

  I know of no previous (theoretical) work that could be used for comparison.

4. **Conclusions**
   - o **First sentence (lines 118-119): Has this been done before? Is this a novel approach? If it has been done before, what is new here? How does it compare to other work?**

     See comments on the novelty of the work above. A sentence has been added to the conclusions, lines 214-215.

- **The section "conclusions" reads more like a "summary", not a conclusion. A conclusion, should ideally go beyond repeating what was observed but give more of an interpretation on a larger scale. What impact do the results have (on clouds, on lab experiments…)? Also an outlook to future work could be given.**

  *Text has been added to the Conclusions, which hopefully improves the section (lines 230-245).*

**Technical corrections:**

- **Theory: Before eq1, give again a reference (Sormakaa and Laaksonen 2007)**

  *Done.*

- **Eq1: S is not introduced in the text above**

  *Corrected.*

- **Before Figure 2 is mentioned, it is not explained what values were chosen for A and B for the calculation of Figure 1.**

  *Corrected.*

- **Line 47: Add "A and B" into the sentence: "Here, the FFH parameters A and B describe …"**

  *Done.*

- **The legend of Figure 1 uses $m_o$ (I guess molecular mass) that has not been introduced and lacks a unit. Also why now here the subscripts "o" instead of "s"?**

  *Corrected.*

- **Figure 1: S\* as a symbol for the critical supersaturation is not introduced in the text or the Figure caption**

  *Corrected.*

- **Caption of Figure 2: instead of using (1), (2), (3), and (4) that can't be found in the Figure, use (black), (blue), (cyan), and (red) and the corresponding descriptions or (solid line), (dashed line), etc.**

  *I have followed the journal instructions here: "A legend should clarify all symbols used and should appear in the figure itself, rather than verbal explanations in the captions (e.g. "dashed line" or "open green circles")." But the numbers are indeed confusing, and I removed them.*

- **Line 108: an "s" is missing. "Figure 4 show[s]…"**

  *Corrected.*

- **Text in figures (axes labels and legend text) is a bit too small**

  *Improved.*

**Reviewer 2:**

The author presents a framework for estimating the cloud droplet activation properties for an insoluble (but water-adsorbing) particle that is coated with a film-forming surfactant. It is pointed out that, because water uptake by such particles is so limited, the particle remains small at its critical diameter and the surfactant can be effective in lowering surface tension. In contrast, strongly hygroscopic particles grow significantly and prior work has shown that the impact of the surfactant can be minimal under high dilution. The conditions under which the surfactant plays a significant role in reducing critical supersaturation for the insoluble particles considered here are mapped out and contrasted with the impacts of instead adding small amounts of hygroscopic material to the particle.

Major Comments: Overall, the manuscript is a nice contribution to the literature on modifications to Kohler theory that consider a broader range of particle types than classic "soluble" and purely "insoluble but wettable" particles. However, it is somewhat difficult to gauge the originality of the contribution based on the prior literature cited. Gohil et al. (2022), not cited in the manuscript, presented the "Hybrid Activity Model" that included the FHH isotherm along with solute water uptake, but they did not include effects on surface tension, as they explicitly acknowledge in their Summary. Therefore, by adding in the equation of state for surfactant films, this manuscript extends their framework and findings. I suggest the author include some additional description along these lines (and compare the two set s derived equations) to better motivate this paper. [Note, There may be other papers since Gohil et al. that have also addressed this topic; I did not check.]

I have added a reference to Gohil et al. (lines 89-91). I do not know of any prior theoretical work that would have considered the impact of film-forming surfactants on the critical supersaturations of insoluble particles.

The non-additivity of effects (lines 136 to end) is an important result. This is discussed here as a specific result for certain parameter choices. It would be very interesting to have more discussion of this point and if possible, to generalize the findings.

Discussion has been added on lines 158-159, 163-168, and 230-236.

.

Minor Comments:

The FHH approach is useful but in practice, are the fit parameters known for atmospherically-relevant particle types? A brief discussion of its practical utility might be helpful.

Nenes and co-workers have determined FHH parameters for a number of atmospherically relevant minerals based on CCN activation experiments, and the FHH parameters of BC particles were discussed extensively in Laaksonen et al. (2020). I have added as to why BC and illite were chosen for this study on lines 47-49.

The partial molecular volume and partial molecular area are identified (lines 72-73) as the relevant parameters for film formation. However, the subsequent discussion and figures sometimes refer to these differently, causing confusion; please make the terminology consistent throughout. Also, on line 76, it's noted that the choices used for illustrative calculations "are not necessarily very realistic". This is unsatisfying; it would be better to provide some estimates of what might be realistic for atmospheric components and use specific examples or a reasonable range for the illustrative calculations.

I have improved the terminology; I now refer to the (partial) molecular area using the term cross-section. The purpose of Fig. 1 is to show schematically how changing the model parameters alter the outcome rather than provide information of atmospherically relevant ranges; I am afraid that if I gave such estimates, they would be based on lab experiments where different surfactants have been used, and I have no idea whether those ranges would be relevant to atmospheric surfactants. So, I hesitate to provide such estimates because I am afraid that someone could actually take them seriously.

Re Figure 1and similar figures: I wondered if presenting as a classic ln S-ln D_dry plot might be helpful? They should appear more linear, and lines of constant kappa can be shown for the solute-only comparisons. However, this is just a suggestion.

I now present the plots in log-log form; interestingly, surfactants make the curves somewhat non-linear.

Gohil, K., Mao, C.-N., Rastogi, D., Peng, C., Tang, M., and Asa-Awuku, A.: Hybrid water adsorption and solubility partitioning for aerosol hygroscopicity and droplet growth, Atmos. Chem. Phys., 22, 12769–12787, https://doi.org/10.5194/acp-22-12769-2022, 2022.